# Qualitative Comparison of 2D and 3D Atmospheric Corrosion Detection Methods

**DOI:** 10.3390/ma14133621

**Published:** 2021-06-29

**Authors:** Thomas De Kerf, Navid Hasheminejad, Johan Blom, Steve Vanlanduit

**Affiliations:** 1Invilab Research Group, Faculty of Applied Engineering, University of Antwerp, 2020 Antwerp, Belgium; steve.vanlanduit@uantwerpen.be; 2EMIB Research Group, Faculty of Applied Engineering, University of Antwerp, 2020 Antwerp, Belgium; johan.blom@uantwerpen.be (N.H.); navid.hasheminejad@uantwerpen.be (J.B.)

**Keywords:** corrosion, confocal laser scanning microscope, image segmentation

## Abstract

In this article, we report the use of a Confocal Laser Scanning Microscope (CLSM) to apply a qualitative assessment of atmospheric corrosion on steel samples. From the CLSM, we obtain high-resolution images, together with a 3D heightmap. The performance of four different segmentation algorithms that use the high-resolution images as input is qualitatively assessed and discussed. A novel 3D segmentation algorithm based on the shape index is presented and compared to the 2D segmentation algorithms. From this analysis, we conclude that there is a significant difference in performance between the 2D segmentation algorithms and that the 3D method can be an added value to the detection of corrosion.

## 1. Introduction

Atmospheric corrosion is the degradation of materials caused by air and the pollutants contained in the air. It can be precisely defined as an electrochemical process that depends upon the presence of electrolytes which may be rain, dew, humidity, or melting snow [1]. The degradation of steel structures due to corrosion is estimated to be 3–4% of a nation’s GDP [2]. The characterization of the corrosion process using microscopic data could lead to a better understanding of the corrosion process, and in turn, lead to a decrease in the overall corrosion cost.

Numerous methods exist to detect or quantify corrosion [3]. They can be categorized into two categories: destructive and non-destructive techniques. Destructive inspection includes corrosion coupons, electrochemical impedance spectroscopy (EIS) [4], electrical resistance (ER) [5], linear polarization resistance (LPR) [6]. Non-destructive methods include ultrasonic measurement, eddy current [7], thermography [8], magnetic flux leakage [9], and image based detection. In this article, we focus on image processing.

There are a variety of image processing algorithms described in the literature to investigate corrosion on metal. Several different approaches exist to achieve corrosion segmentation, and most algorithms can be classified into four categories:Statistical analysis: Using first-order statistical parameters, such as mean, skewness, variance, kurtosis, energy and entropy to describe corrosion patterns [10,11,12];Color space transformations: Converting the images to a different color space in order to facilitate segmentation [13,14,15];Transform based models: Using methods such as Fourier transformations [16,17,18] or wavelet decomposition [19,20,21,22,23] to aid segmentation;Co-occurrence matrix: Calculating several properties of the Gray-Level-Co-Occurrence Matrix (GLCM) [24], such as energy, entropy, contrast, dissimilarity and homogeneity could provide more textural details of the corrosion pattern [25,26,27];Supervised Machine Learning: Using a database with labeled corrosion/non-corrosion samples, machine learning or neural networks can be used to segment images. Examples can be found in [28,29,30,31].

Note that only unsupervised algorithms are discussed in this article. Supervised algorithms will not be taken into account for the comparison. It is common for corrosion detection algorithms to use a combination of different algorithms to process images. For instance, in [18], a variety of color space transformations and transform-based models (Fourier transformation) are used to process the images.

The previously described image processing techniques have the advantage of being fast and are a low-cost way of analyzing large amounts of data. However, the algorithms could be prone to false positives [13,32]. For instance, if the entire image is covered in corrosion products, it will not be straightforward to correctly identify corrosion due to the absence of a clean part. Uneven illumination or a failure to distinguish background from corrosion products can also be detrimental to the segmentation performance.

To resolve these issues, there have been investigations to use surface height data to characterize corrosion effects. The height profile adds a physical significance to the detection algorithm. For instance, ref. [33] uses standard statistical surface metrology methods to identify pitting corrosion in aluminum plates. Atmospheric corrosion in carbon steel materials is not investigated.

This article presents a novel segmentation method that uses the data from the 3D height profile. This segmentation method combines several often used algorithms, such as color space transformations, co-occurrence matrix, and statistical analysis. These well-known algorithms are combined with a surface shape parameter as defined in [34].

The structure of this article is as follows: in Section 2, the experimental parameters are illustrated, the different 2D algorithms are summarized, and the innovative 3D segmentation algorithm is introduced. This is followed by Section 3, where firstly, the results of the 2D algorithms are compared to each other. Secondly, the best 2D method is compared to the proposed 3D method. In Section 4, the conclusion is presented, and possible further investigations are proposed.

## 2. Material and Methods

In the following section, the preparation of the corrosion samples is explained, the parameters of the microscope are described, and the used algorithms are presented in detail.

### 2.1. Corrosion Samples

Five samples were created that underwent cleaning and, subsequently, environmental corrosion exposure. The initial sample was a plate of S235 carbon steel of 200 mm by 50 mm and a thickness of 5 mm. In this plate, five squares were cleaned through laser ablation. These squares are 30 by 30 mm in size and done with a robot arm to minimize variations within a single square. The plate was placed outside for five days to undergo atmospheric corrosion to create a realistic corrosion surface. After this step, the samples were cut into 50 by 50 mm parts to simplify the measurement under the microscope.

### 2.2. Confocal Laser Scanning Microscope (CLSM)

The confocal laser scanning microscope used in this research is a Keyence VK-X1000. This equipment allows us to have an output RGB image and a heightmap from the same surface area to compare 2D and 3D imaging methods. The preprocessing was performed with the Keyence MultiFileAnalyzer software. Using this software, a leveling of the reference plane is done to account for the tilt between the sample and the microscope base. After this preprocessing step, the image data and the height data are exported, and further analysis is performed in Python.

### 2.3. Corrosion Segmentation Methods

In this article, four 2D segmentation methods and one 3D method are implemented to segment each pixel from the RGB images into corrosion and non-corrosion classes. These methods are:Choi and Kim algorithm (2D) [12];Shen algorithm (2D) [18];Medeiros algorithm (2D) [25];Ghanta algorithm (2D) [23];Shape Index algorithm (3D).

The goal was to stay as close as possible to the algorithms described in the original articles, but some modifications were inevitable. Differences between the original article and the implementation in this research are described in the following section, along with a schematic of the algorithm.

#### 2.3.1. Method 1—Choi and Kim Algorithm

The algorithm presented by Choi and Kim [12] uses a combination of statistical parameters (mean, median, and skewness) and texture parameters obtained from the co-occurrence matrix. As a first step, the RGB image is converted to the HSI color space. The statistical parameters are calculated on squares of 10 pixels by 10 pixels. In order to calculate the texture parameters, a sliding window (11 pixels by 11 pixels ) is moved over the image. In each iteration, the co-occurrence matrix is calculated. Based on these co-occurrence matrices, the following texture parameters are calculated: contrast, dissimilarity, homogeneity, energy, correlation, and angular second moment. These statistical and texture parameters are then joined into a single matrix, and a k-means clustering algorithm is applied to segment the images into two categories. This last step is a deviation from the original article where a linear regression model is used. The unsupervised k-means clustering algorithm was necessary to obtain a pixel-wise segmentation of the image. A graphical overview of the algorithm, as implemented in this paper, can be found in Figure 1.

#### 2.3.2. Method 2—Shen Algorithm

In the article by Shen et al. [18], an automated corrosion detection algorithm is presented that uses a filter approach in the frequency spectrum. A large selection of color spaces and filter parameters are evaluated to present the best parameters to assess corrosion in RGB images. A graphical representation of the different algorithm steps can be found in Figure 2. First, the image is converted from RGB to CMY color space. A Fourier transformation is then applied to the image. This frequency-based representation of the image is then multiplied by an ideal low pass filter (ILPF) with a D_0_ of 60%. The next step is to perform an inverse Fourier transform and convert it back to RGB color space. From this filtered RGB image, each pixel is compared to a predefined rust spectrum with following parameters:Center (R,G,B) = (193.54, 124.25, 59.61);Standard deviation (R,G,B) = (29.95, 30.15, 22.13).

#### 2.3.3. Method 3—Medeiros Algorithm

The third corrosion segmentation method is based on the article by Medeiros et al. [25]. In this article, statistical moments are combined with gray-level co-occurrence features. As a first step, the image is split into squares of 128 pixels by 128 pixels. These patches undergo two different paths: a co-occurrence path and a statistical path, before being combined, as illustrated in Figure 3. In the co-occurrence path, the RGB image is converted to grayscale, and the Gray-Level-Co-occurrence Matrix (GLCM) is calculated with the following parameters: contrast, correlation, energy, and homogeneity. For the statistical path, the image is converted into HSI color space, and four statistical moments are calculated for each component (H, S, I). Those two paths combined form a matrix comprising 8 features. This number is reduced through Principal Component Analysis (PCA) to 5 features. A k-means clustering algorithm is applied to segment the patches into corrosion and non-corrosion classes from these five features. The difference between the original paper and the algorithm utilized in this paper is the size of the patches (128 pixels in the original versus 32 pixels) and the clustering algorithm. In the original article, a Linear Discriminant Analysis (LDA) is performed on the dataset. However, an unsupervised clustering algorithm such as k-means clustering is chosen for easy comparison between the different algorithms.

If a pixel is inside the range of the center ± standard deviation, it is classified as corrosion. This algorithm is not modified in regards to the original paper.

#### 2.3.4. Method 4—Ghanta Algorithm

In the article by Ghanta et al. [23], a combination of wavelet transformation and intensity averaging is used to detect corrosion in RGB images. In Figure 4, an overview of the algorithm is presented. First, the image is split into 8 pixels by 8 pixels patches. These patches follow two paths: a wavelet path and an average intensity path. In the wavelet path, one level of wavelet transformation is applied to the patch for every component (R, G, B). The result of this transformation is four sub-band images. The energy and entropy are calculated for each of these sub-band images. This results in a matrix with 24 features: three (R, G, B) by four (wavelet transformations) by two (energy and entropy). In the other path, we convert the RGB image into YIQ color space and calculate the mean value of the I band. This feature is combined with the other 24. These 25 features are reduced to five by applying a PCA algorithm. On this matrix with five features, a k-means clustering algorithm is applied. The difference between the original paper and this implementation is the use of the clustering algorithm. In the original paper, a least mean squares classifier is used to classify the patches. However, as mentioned before, an unsupervised clustering algorithm is used in this study for the sake of comparison.

#### 2.3.5. Proposed 3D Height Segmentation Method

The algorithm to segment the 3D height data into corrosion non-corrosion classes combines the previous 2D image segmentation algorithms. An overview of the algorithm steps is presented in Figure 5. First, a denoising filter [35] is applied to the data. This denoised data follow two paths, a shape index path and a combined statistical and texture parameters path. The shape index is a parameter that classifies specific height profiles in different categories based on the shape of the 3D data, see [34]. In the other path, a sliding window is moved over the image and, for every window, the following parameters are calculated: mean, median, skewness, similarity and correlation. These parameters are then combined with the shape index, and a PCA is applied to reduce the number of features. Finally, on these remaining features, a k-means clustering algorithm is applied to segment the image.

## 3. Results

The results are presented in two sections. The first part, Section 3.1, deals with the results of the 2D corrosion segmentation methods. This section presents a global overview of these algorithms, together with a close-up of specific corrosion spots. These spots are chosen because there is a clear difference between the results of the different algorithms. The second part, Section 3.2, shows the results of the 3D segmentation methods compared to the best performing 2D segmentation algorithm. As with the previous part, a general overview is first presented, and then, specific close-ups of the samples are discussed.

### 3.1. 2D Segmentation Algorithms

#### 3.1.1. General Overview

Figure 6 presents an overview of the segmentations performed by the different algorithms together with the original RGB image. This overview shows that there are significant differences between the output of the algorithms used in this article. The results of methods 1 and 2 are comparable. The resulting segmentation map has less noise than methods 3 and 4. In Sample 5, method 2 is slightly overestimating the amount of corrosion, whereas method 1 does not. Using method 3, the corrosion patches are still present, although they are less pronounced because there are many wrongly classified pixels. These erroneous pixels causes a darker appearance. A remarkable effect is present in Sample 1, method 3. There are several horizontal stripes in the segmented image. These stripes only occur in method 3. Method 4 appears to overestimate the amount of corrosion in every image. Not only are the corrosion patches that are present in the RGB image exaggerated, but in some samples, this method seems to create new patches. This is very visible in Sample 5, where the top right corner is classified as corrosion, yet this is not seen in the RGB image.

#### 3.1.2. Close-Up at Problem Spots

Figure 7 shows a large corrosion spot in sample one. The corrosion spot is visible in the optical image. When looking at the four algorithms, methods 1, 2 and 4 perform equally good. Method 3 introduces a significant amount of noise in the segmented image. Varying the size of the image patches did not affect the performance of this algorithm. However, additional noise suppressing algorithms could be beneficial. Method 4 is slightly overestimating the amount of corrosion, but the overall shape is still very accurate.

Figure 8 is an extract of the top region of Sample 5. In this image, several metallic clean spots are visible, along with corrosion spots. Method 1 is considerably accurate, but some small metallic parts are classified as corrosion (see red arrow in Figure 8). Method 2, however, shows a low accuracy in predicting the right half of the image. Almost the entire right half is classified as corrosion, whereas only a small portion is visible in the RGB image. Method 3 suffers from the noise, and several metallic spots are wrongly classified as corrosion. Method 4 is overpredicting the amount of corrosion in the image. It appears that the clean metal parts are correctly identified, but the corrosion and gray metallic surfaces are both classified as corrosion. This could be improved by increasing the number of clusters in the clustering algorithm.

#### 3.1.3. Summary of 2D Algorithms

From the general overview as seen in Figure 6 and the close-ups, it is shown that the Choi and Kim [12] algorithm is the best suited for corrosion detection for this specific dataset. The other algorithms have shown their merit for other datasets. None of these algorithms used microscopic images to classify corrosion, explaining the difference in prediction quality between the algorithms. The goal of this article was to stay as close as possible to the original article, which means no extra fine-tuning of the algorithms was realized. This could be a source of improvement for the lesser performing algorithms.

### 3.2. 3D Segmentation

#### 3.2.1. General Overview

Figure 9 shows the original RGB image, raw 3D height profile and the result of the 3D segmentation method, as introduced in Section 2.3.5. When observing the 3D height column, it is apparent that there are corresponding peaks and dales visible for every significant corrosion spot. However, for more minor corrosion spots, there are no significant hills or dales. It appears that the corrosion spot has to be at least of a specific size to show up in the height profile clearly. Sample 5 is an exception in displaying the corrosion spots. Only one high peak is displayed very prominently, and the other corrosion spots are underestimated. When looking at the column with the 3D segmentation method applied, Samples 1 and 2 are segmented with very little noise added to the image. However, in Samples 3 and 4, there is a significant amount of noise introduced. The major corrosion spots are still clearly detected, but the corrosion segmentation is less precise for small spots.

#### 3.2.2. Closer Look at Problem Spots

In this section, two close-ups of the samples are chosen to identify the difference between the 3D segmentation algorithms and the best performing 2D segmentation algorithm. In addition, several areas were chosen where there is a clear difference between the two segmentation algorithms. The merits of both methods will be discussed. In Figure 10, a close-up of Sample 3 is presented, together with the 2D and 3D segmentation. There is a significant shape difference between the two methods. The 2D method appears to be performing well, and the 3D method underestimates the amount of corrosion. In the 3D method, all of the corrosion clusters are present but less profound, and they can be mistaken for noise.

In Figure 11, a close-up of Sample 5 is presented. There is a clear difference between the two segmentation methods. Here, the 2D method is overestimating the corrosion spots. Some spots that are clean metal are segmented as corrosion spots. The 3D method does not show these false positives. However, the corrosion is overall underestimated.

## 4. Conclusions

From the general overview and the close-up of the corrosion spots, a significant variation in performance between the different 2D algorithms is observed. Method 1 (Choi and Kim) and Method 2 (Shen) appear to be generating the most accurate results. Method 3 (Medeiros) introduces a significant amount of noise in the segmentation image. Method 4 (Ghanta) is overall the worst performing of the four. This method is not suited to detect corrosion from microscopic images accurately. The 3D segmentation algorithm can correctly identify the corrosion clusters. However, it consistently underestimates the amount of corrosion present. With the 3D segmentation method, the effect of corrosion (dales) and corrosion products (hills) will be measured, thus introducing a physical significance to the corrosion detection. In comparison, the 2D method is better at classifying the correct shape for the corrosion spots. Therefore, a combination of the 2D and 3D segmentation methods coupled with a decision algorithm based on the size of the corrosion spot can be the topic of future research to improve the overall accuracy.

## Figures and Tables

**Figure 1 materials-14-03621-f001:**
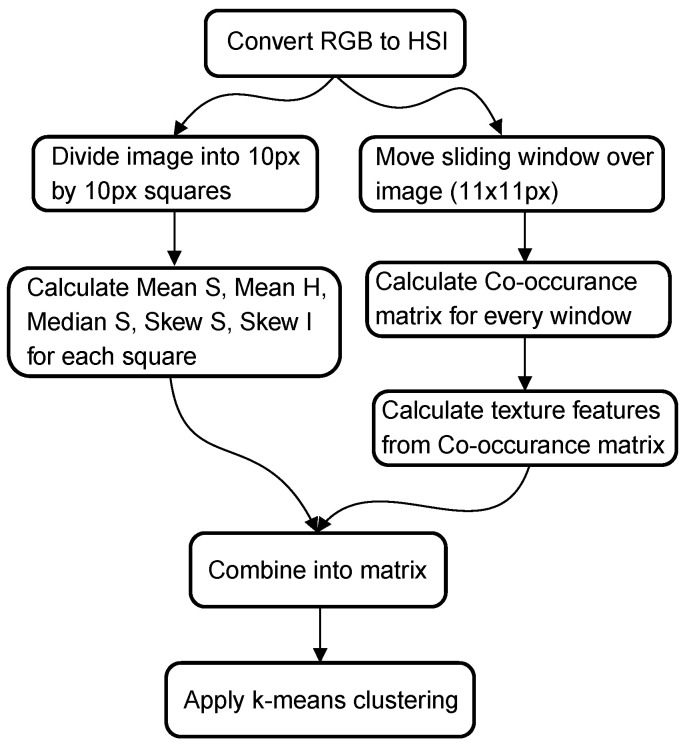
Graphical overview of method 1 based on Choi and Kim [12].

**Figure 2 materials-14-03621-f002:**
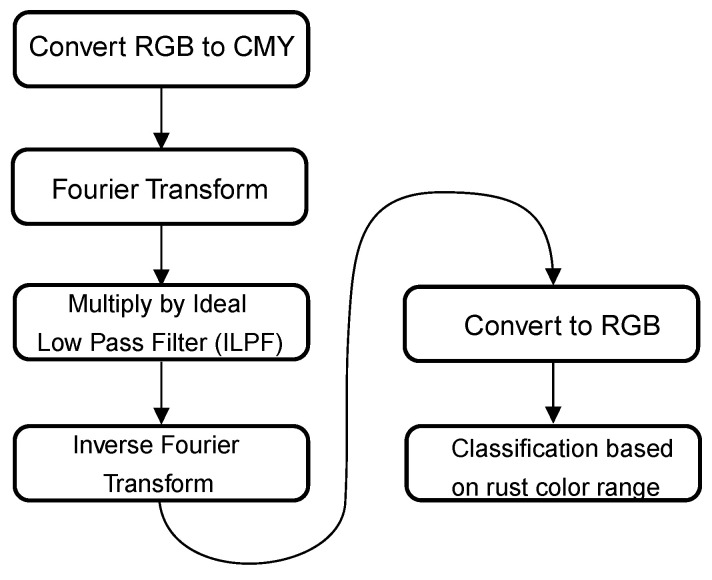
Graphical overview of method 2.

**Figure 3 materials-14-03621-f003:**
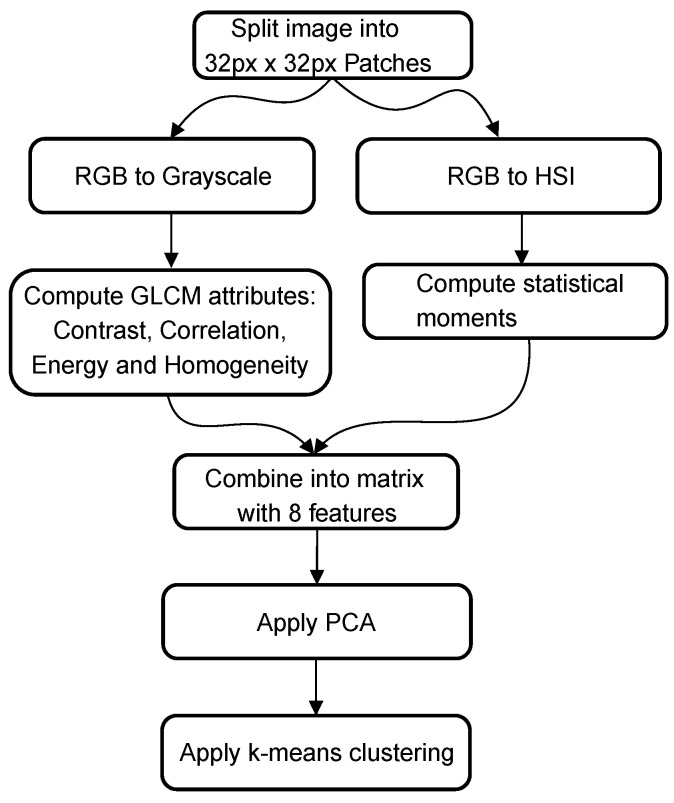
Graphical overview of method 3.

**Figure 4 materials-14-03621-f004:**
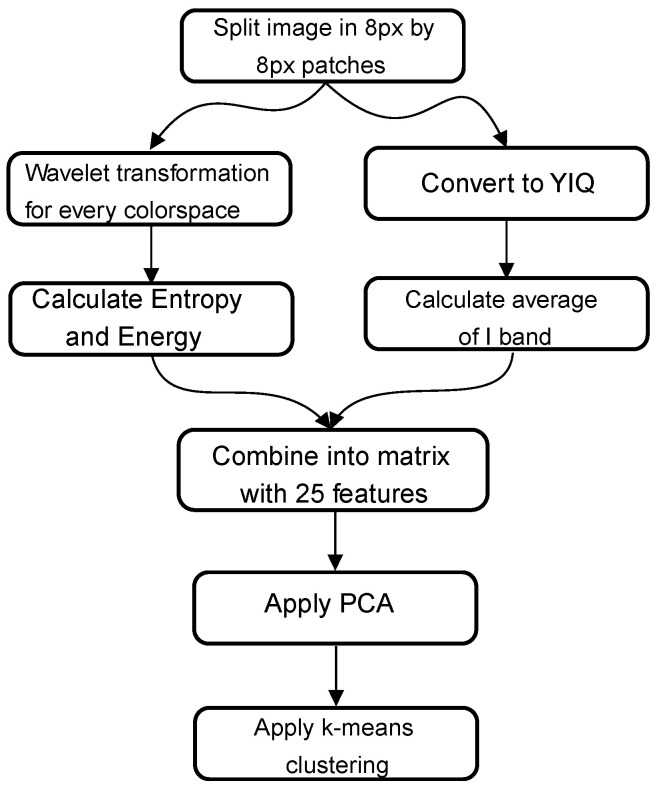
Graphical overview of method 4.

**Figure 5 materials-14-03621-f005:**
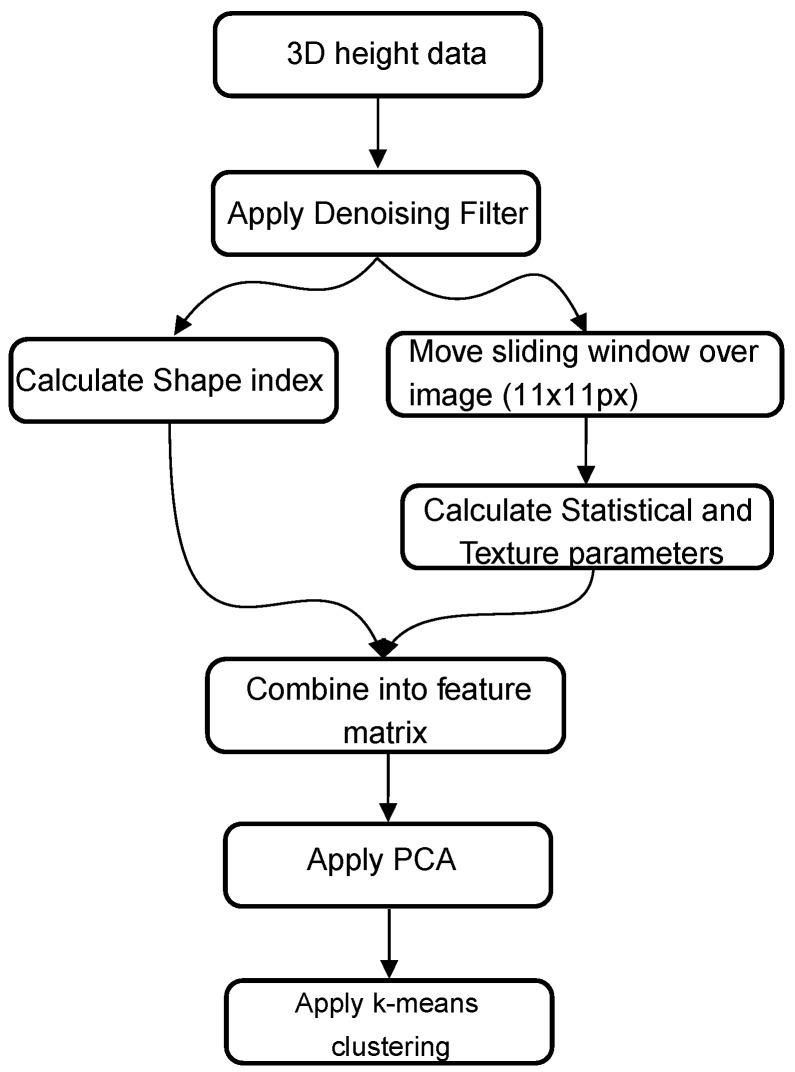
Graphical overview of the segmentation algorithm for 3D height data.

**Figure 6 materials-14-03621-f006:**
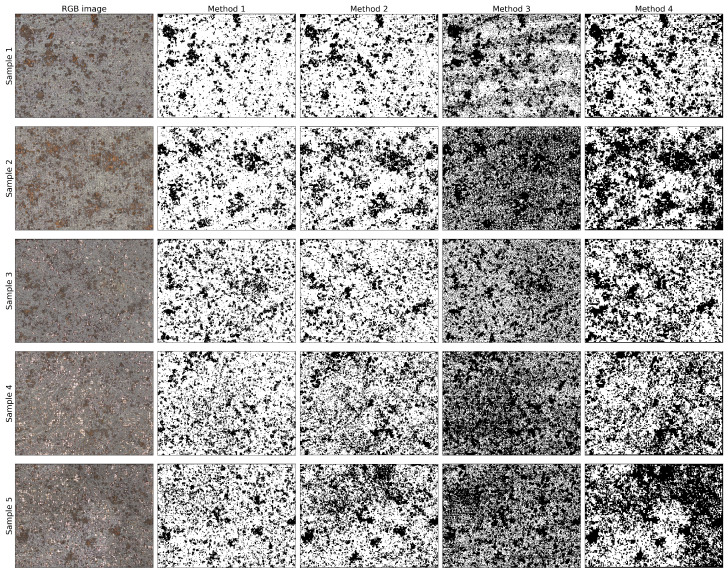
Overview of different segmentation methods for the investigated samples. Each row represents a different sample, and every column shows a distinct analysis method.

**Figure 7 materials-14-03621-f007:**
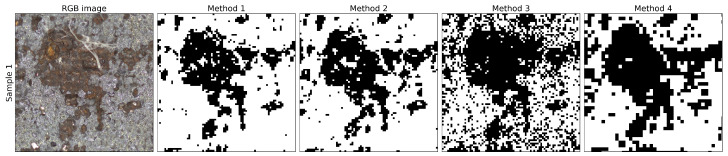
Close-up of a corrosion spot in sample one.

**Figure 8 materials-14-03621-f008:**
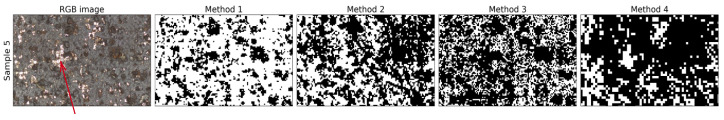
Close-up of a corrosion spot in Sample 5, the red arrow indicates a clean and bright metallic patch.

**Figure 9 materials-14-03621-f009:**
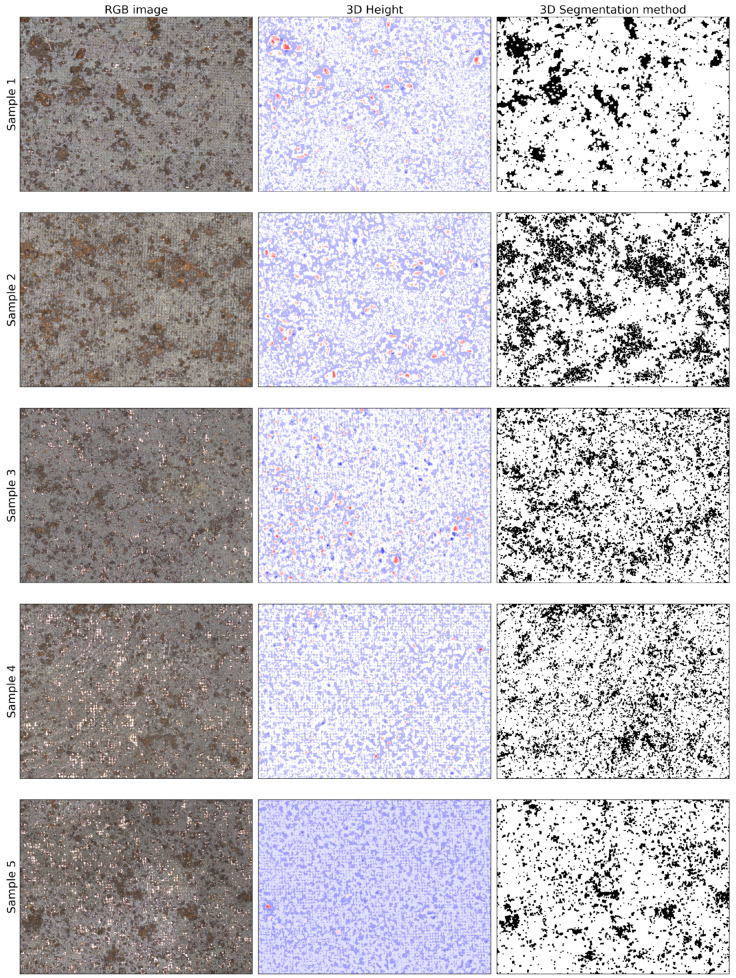
Overview of different segmentation methods based on the 3D height data for the investigated samples. Each row represents a different sample. The first column is the RGB images, the second column displays the 3D height data, and the third column shows the segmentation result.

**Figure 10 materials-14-03621-f010:**
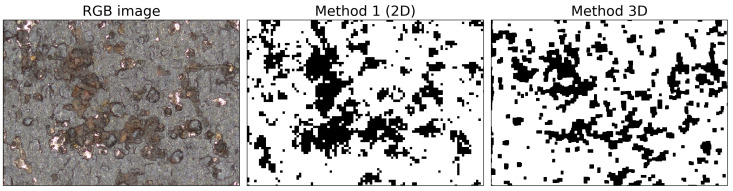
Close-up of Sample 3. The first image is the RGB image, the second image is the 2D segmentation (method 1), and the third image is the result of the 3D segmentation method.

**Figure 11 materials-14-03621-f011:**
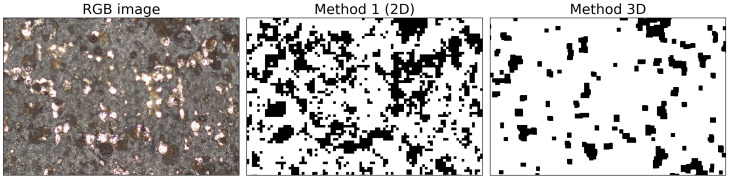
Close-up of Sample 5. The first image is the RGB image, the second image is the 2D segmentation (method 1), and the third image is the result of the 3D segmentation method.

## Data Availability

Data is contained within the article.

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
