# Peer review of "Qualitative Comparison of 2D and 3D Atmospheric Corrosion Detection Methods"

_materials, 2021, doi:10.3390/ma14133621_

Round 1

Reviewer 1 Report

Good paper, well written, exhausting and detailed 2D and 3D analysis of carbon steel coupons after atmospheric corrosion. One team extensively verified four 2D analytical methods and one 3D analytical method. All experiments were done with the same coupons, so obtained results are convincing and clearly presented. Congratulations to Choi and Kim [Ref.12] authors of the best analytical method. I will strongly urge the PT Authors to continue their research with other coupons (carbon steel, Al) in correlation also with other no optical corrosion detection method (eg EIS analysis).

Small remarks - page 1 line 26 - ther are discussed four or five categories ?

Page 3 line 92 - why "Algorithm" while in all other methods there was used term "algorithm"

There are two main objectives discussed within this paper: a) comparison of four 2D microscopic observation analytical methods of steel coupons subjected to the corrosion environment b) comparing these results of these four methods 2D observations with 3D confocal microscopic observations.

Studying the corrosion processes and mostly prevention of corrosion with different chemical compounds forming monolayers and well adhering to the metal surface this paper is interesting because makes a comparison between four 2D microscopic observation methods. In the research somebody can use this one or another 2D inspection methods and this paper gives an importnat statement: method 1 and 2 give better results than method 3 and 4. This is why this paper is valuable for me - gives direct indication - use this and this 2D inspection method 

Comaprison between four published 2D inspection methods is important but from the scientific point of view, searching for novelty, is not so much important. Analysis of 3D inspection method with confocal microscope is a new method but this is rather "broadening" the reserach scope than going into deeper analysis. Therefore I have suggested to combine these microscopic observations with other corrosion inspection method for example EIS analysis 

The paper is clearly written, but I cannot say something about English language. The text is clear, added figures clarifies discussed topics 

Conclusions to presented data are clear but will be better if conclusions cover also other analytical method like eg EIS analysis.

Author Response

Dear Reviewer,

Thank you for taking the time to review our submitted article. 
I will address the comments in the following matter, firstly the original comment, then the reply, in blue.

Small remarks - page 1 line 26 - there are discussed four or five categories ?
This was indeed a typo; there are five categories. The text has been changed to five.

Page 3 line 92 - why "Algorithm" while in all other methods there was used term "algorithm"
I corrected the line 92 Algorithm to algorithm, to improve consistency.

Comaprison between four published 2D inspection methods is important but from the scientific point of view, searching for novelty, is not so much important. Analysis of 3D inspection method with confocal microscope is a new method but this is rather "broadening" the reserach scope than going into deeper analysis. Therefore I have suggested to combine these microscopic observations with other corrosion inspection method for example EIS analysis

This is a valid remark; a more 'chemical' ground truth map of corrosion would be beneficial to apply this comparison. However, since EIS only provides us with a corrosion rate for a (larger) area, this is not suited to compare the techniques used in this paper.  

Reviewer 2 Report

This manuscript performed a comparison study among different algorithms to analyze the atmospheric corrosion on steel samples. The authors have created five samples and collected images with CLSM. The difference resulted of four 2D and one 3D methods were compared. The results were clearly presented and future improvements have been suggested. Although the proposed 3D method does not improve in estimating the amount of corrosion, it introduces a physical significance to the corrosion detection. The conclusion should benefit the community and provide potentially useful information. I have no problem of recommending publication of this manuscript.

However, there are some places that can be further improved in the current manuscript.
1. In order to compare the performance among different algorithms, the authors presented some images in Fig.6-10 from which a rough and intuitive comparison can be obtained. However, further analysis is missing and there is not a quantitative discussion about the performance of these methods.In my opinion, insteading of just saying one method is better than others, it may be more helpful to readers if the authors could find some index or quantity from which an overall measurement of the performance can be made. 
2. The introduction of the height profile into the corrosion analysis is useful and the peaks and dales can be observed corresponding to the significant corrosion spots. However, this fails in the detection of small spots due to their limited sizes. Can the authors make comments about how this knowledge may be helpful to eliminate the false positives in the 2D analysis?
3. The 2D method performed better in sample 3 but worse in sample 5 than the 3D method. What are the underlying reasons for this difference and how could this help us determine which method should be used in practice?

Above are my suggestions or minor comments on the current manuscript.  This paper is well written, and the topic is original and interesting.
The main question has been well addressed and the conclusion is straightforward although an advanced description/analysis may be helpful to make it  more quantitative.
Therefore, I recommend this paper for publication after a minor revision.

Author Response

Dear Reviewer,

Thank you for taking the time to review our submitted article. 
I will address the comments in the following matter, firstly the original comment, then the reply, in blue.

  1. In order to compare the performance among different algorithms, the authors presented some images in Fig.6-10 from which a rough and intuitive comparison can be obtained. However, further analysis is missing and there is not a quantitative discussion about the performance of these methods. In my opinion, instead of just saying one method is better than others, it may be more helpful to readers if the authors could find some index or quantity from which an overall measurement of the performance can be made.
    A comparison metric would indeed be very interesting to be able to compare the different methods quantitatively. However, finding this ground truth from RGB images is very subjective. Therefore, the authors did not implement a comparison metric in the article
  2. The introduction of the height profile into the corrosion analysis is useful and the peaks and dales can be observed corresponding to the significant corrosion spots. However, this fails in the detection of small spots due to their limited sizes. Can the authors make comments about how this knowledge may be helpful to eliminate the false positives in the 2D analysis?

    As the results show that the 2D method is very accurate in detecting the small samples and the 3D method is more accurate for the larger corrosion spots, a dual approach could be implemented. Using a threshold on the diameter size of the corrosion spot, we could filter out the large spots with the 3D method and the smaller spots with the 2D method. And as the last step, combine the 2D and 3D corrosion maps.

    I have included the following text in the final paper:

    Therefore, a combination of the 2D and 3D segmentation method coupled with a decision algorithm based on the size of the corrosion spot can be the topic of future research to improve the overall accuracy.

  3.  The 2D method performed better in sample 3 but worse in sample 5 than the 3D method. What are the underlying reasons for this difference and how could this help us determine which method should be used in practice?
    The difference between sample 5 and sample 3 is that there are many more clean, metallic parts in sample 5. These metallic parts tend to be classified as corrosion spots with the current 2D method. The overall cleanliness or background color will determine the accuracy of the 2D method. The background color or cleanliness will have no influence on the accuracy of the 3D method.
